# Enzymatic Activity of Soil after Applications Distillery Stillage

Agata Bartkowiak *, Joanna Lemanowicz , Magdalena Rydlewska, Olga Drabińska and Karol Ewert

Department of Biogeochemistry and Soil Science, Faculty of Agriculture and Biotechnology, Bydgoszcz University of Science and Technology, 6/8 Bernardyńska Street, 85-029 Bydgoszcz, Poland; jl09@interia.pl (J.L.); magryd@pbs.edu.pl (M.R.); olga51.bydgoszcz@wp.pl (O.D.); nokiakarolek@wp.pl (K.E.)
* Correspondence: bartkowiak@utp.edu.pl; Tel.: +48-52-374-95-26

**Abstract:** This study aimed to evaluate the fertilizing value of rye stillage used in the cultivation of winter triticale cv. 'Grenado'. The research was performed in 2018 (autumn, before the application of the stillage), 2019, and 2020 (spring and autumn after stillage application) on Luvisoil at the depth levels of 0–20 and 20–40 cm. Each year, the basic soil parameters were analyzed, i.e., pH in 1M KCl, organic carbon (Corg), available phosphorus (P), potassium (K), magnesium (Mg), zinc (Zn), and copper (Cu). Enzymes were also accounted: alkaline phosphatase (AlP), acid phosphatase (AcP), and dehydrogenases (DEH). The use of stillage resulted in a significant increase in the content of P, K, and Mg and the activity of AlP, AcP, and DEH in the soil. It significantly increased the Corg content and did not have a significant effect on pH. The obtained results indicate that the response of the enzymatic activity to the distillery stillage depended on both the sampling season soil and the depth. However, it is necessary to systematically monitor the pH of the soil and at the same time to rationally apply mineral fertilization.

**Keywords:** alkaline and acid phosphatase; dehydrogenases; distillery stillage; Luvisoils; macro- and microelements

## 1. Introduction

Organic waste plays an increasingly important role in soil fertilization. Currently, due to the steady decline in the size of animal husbandry in Poland and a decrease in the amount of manure application, increasing opportunities are facing the use of organic waste produced by the agri-food industry. Distillery stillage is an example of such waste, which, under the Regulation of the Minister of Environment, Natural Resources, and Forestry of 24 December 1997, is classified as agri-food waste [1]. This interest in stillage waste is caused by the increasing number of distilleries and the related amount of waste produced. Distillery waste is one of the most polluting waste products due to its low pH, high temperature, and high content of organic matter [2].

Stillage is a by-product of a distillery obtained from the processing of starchy products, such as grains, potatoes, and molasses. It is produced with spirit by distilling fermented mash. The quantity and quality of the obtained distillery stillage are highly diverse, as it depends on the type and quality of the plant material it comes from, as well as on the course of the fermentation process, technology, and equipment used. The main raw materials used for the production of spirit in Polish distilleries are cereals (rye, wheat, maize) and potatoes. The most popular stillage is of grain origin, which accounts for ca. 74% of the total stock produced, and the least popular is molasses—accounting for 3.2% [3]. The surplus of stillage production is a problem for distilleries, which are forced to look for its recipients. Therefore, the possibility of using stillage for fertilization purposes deserves special attention. This has been successfully practiced for many years in Poland and other countries around the world [4–7]. Distillery stillage contains, along with organic carbon compounds, mineral nutrients essential for plants, which allows its use through the R10 recovery process—spreading on the soil surface to fertilize or improve the soil [8]. They

are characterized by a high content of potassium, calcium, magnesium, and sulfur. The common feature of stillages is the low phosphorus content compared to nitrogen and potassium and a relatively high content of organic carbon, mainly in the form of organic acids [4,9]. A limitation in the usage of stillage can also be its low pH. This waste is acidic and tends to be extremely acidic as fermentation progresses. Organic acids present in the stillage, responsible for the low pH, undergo biodegradation in soils, the end products of which are $CO_2$ and $H_2O$ [10–12]. The disadvantage of distillery stillages is their high hydration, which means that when used for agricultural purposes they are often treated as wastewater. The use of stillage for fertilization can bring benefits to the agricultural sector, as it not only enables the recycling of minerals, but can also reduce the costs associated with the use of mineral fertilizers. The distillery stillage can improve the fertility of the soil by increasing its abundance in humus, nitrogen, and potassium, and reducing its acidification [13–15]. Gahlota et al. [16] reported that the improper use of stillage can adversely affect soil properties by lowering the level of plant nutrients or increasing soil salinity. Nandy et al. [17], on the other hand, observed an increase in the number of microorganisms after the application of rice stillage.

Despite the growing interest in stillage application, the processes and mechanisms of its long-term impact on the enzymatic activity of soil have not yet been sufficiently understood. Soil enzymes are a sensitive indicator used in the study of the relationship between plants and soil [18]. Enzymes are biological catalysts related to the cycle of macronutrients (C, N, P, S). The soil enzyme activity results provide information on biochemical processes. In the assessment of agricultural soil, in addition to physical and chemical parameters, the activity of selected enzymes, which have been recognized as indicators of soil quality, is determined [19]. The use of enzymatic activity as markers for the assessment of soil fertility enables the quantification of anthropogenic changes and allows for the monitoring and identification of trends occurring in the soil [18]. According to Kaur et al. [20] and Hassan et al. [21], the studied soil enzymes, participating in the circulation of nutrients in the soil, showed much higher activity with an increase in the concentration of the stillage used. The contradictory results of the research contributed to the analysis of the influence of rye stillage on the cultivation of winter triticale.

The aim of the study was to determine the effect of the use of stillage under the cultivation of winter triticale cultivated in a long-term monoculture on the enzymatic activity of soil and its relations with selected chemical parameters during the three-year research period.

## 2. Materials and Methods

### 2.1. Location of Soil Sampling

The research areas were located in the intensively used agricultural region of the Kuyavian-Pomeranian Voivodeship ($52°49′02″$ N $17°34′25″$ E). The area included in the analysis is located in the eastern part of the Chodziejski Lake District, located between the valley of the central Noteć River, and the valley of Wełna, the right tributary of the Warta River (north-central Poland). The area receives 512 mm of rainfall per year (mainly in the summer months) and the average annual temperature is 8.1–8.6 °C. Luvisoils are the dominanting type of soils on the surveyed area [22]. In spring (April), pre-sowing fertilization was made in the quantity: $P_2O_5$ 30 kg ha$^{-1}$, $K_2O$ 50 kg ha$^{-1}$, and N 30 kg ha$^{-1}$. Soil samples were collected from surface mineral levels from depths of 0–20 cm and 20–40 cm. At each period, 40 soil samples were collected (20 soil samples in the depths 0–20 cm and 20 soil samples in the depths 20–40 cm. Soil samples were collected by the squares method from the plot of 2 ha. The average sample consisted of ten primary samples. The distillery stillage was applied in October 2018 at a dose of 40,000 L per hectare for the cultivation of winter triticale cv. 'Grenado' in monoculture. Soil samples were taken before the use of the stillage (control in 2018) and seasonally (spring, autumn) in the following two years (2019 and 2020) after applying the stillage as a fertilizer. The chemical composition

of the studied distillery stillage produced during the processing of rye mainly is given in Table 1.

**Table 1.** Characteristics of distillery stillage.

| pH | N | P | K | Mg | Zn | Cu |
|----|----|----|----|----|----|----|
| | | | g L$^{-1}$ | | | mg L$^{-1}$ |
| 3.8 | 2.56 | 1.38 | 12.6 | 1.95 | 4.21 | 0.76 |

*2.2. Soil Analysis*

Chemical analyses were performed on air-dried and sieved samples (<2 mm). Each sample was analyzed in triplicate. In the properly prepared soil samples, the following parameters were assayed:

- Soil pH in 1 M KCl [23].
- Organic carbon (Corg) was determined using Tiurin's method by wet oxidation at 180 °C with a mixture of potassium dichromate and sulfuric acid [24].
- The granulometric composition with the laser diffraction method applying the Masterssizer MS 2000 analyzer.
- The contents of available forms of phosphorus (P) [25] and potassium (K) were defined, by the Egner-Riehm method (DL) [26], as was the content of magnesium available to plants (Mg) following the Schachtschabel method [27].
- The contents of easily available forms of heavy metals (Zn, Cu), DTPA-extracted (1 M diethylenetrianinepentaacetic acid) were also measured, according to Lindsay and Norvell [28]. The content of the available forms of Zn and Cu to soil was determined by atomic absorption spectroscopy using a Solaar S4 spectrometer. To verify the accuracy of the results, the analysis of the certified material Loam Soil No. ERM-CC141 and the so-called zero tests were carried out, which were exposed to the identical analytical procedure as the soil samples. Good comparability between the certified and determined values was obtained.

*2.3. Enzyme Analysis*

The activity of the surveyed enzymes was studied in fresh, humid, and sieved (<2 mm) soils that were stored at 4 °C for two weeks. Each activity test was repeated three times.

- Dehydrogenases (DEH) activity was determined by Thalmann's method [29] after incubating the samples with 2,3,5-triphenyltetrazolium chloride and measuring the absorbance of triphenylformazane (TPF) at 546 nm and expressed as mg TPF kg$^{-1}$ 24 h$^{-1}$.
- The activity of alkaline phosphatase (AlP) and acidic phosphatase (AcP) in the soil was measured based on *p*-nitrophenol detection (pNP) produced after incubation (at 37 °C, for 1H) at pH~6.5 for AcP and pH~11.0 for AlP [30].

*2.4. Statistical Analysis*

The results were analyzed statistically using STATISTICA 13.0 software. All analytical measurements were performed in three replicates. Data on all soil parameters were subjected to a Shapiro–Wilk test to check the normality of empirical distributions.

The results were analyzed for simple correlation ($p < 0.05$) that determined the degree of dependence between respective features. Data from Corg measurements, available forms of P, K, Mg, Zn, Cu, and enzymatic activity (AlP, AcP, and DEH) were analyzed using a two-way analysis of variance (ANOVA), where the first factor was the soil sampling season (2018 autumn, 2019 and 2020 spring, autumn) and the second factor was the depth of sampling. To study the tendency (mean, median) and variability (standard deviation SD, minimum and maximum) of the sample population, classical statistics were used. The coefficient of variation (CV%) for the analyzed parameters was also calculated. The CV values of

0–15%, 16–35%, and >36% mean low, moderate, or high variability, respectively [31]. The multidimensional exploration technique of the main components of PCA was used to explain the differentiation of the soil in relation to the studied enzymes (AlP, AcP, and DEH) and the content of clay, Corg, P, K, Mg, Zn, Cu, and pH in KCl before and after the application of the decoction distillery in terms of the first two components.

## 3. Results and Discussion

Table 2 presents the results of selected physical and chemical properties and enzymatic activity of the soil before the application of stillage. As the soil texture is a parameter that does not undergo seasonal changes, the content of the loam fraction is presented only in Table 2. It remained unchanged for the rest of the time. In the research of the granulometric composition, it was found that the analyzed soil samples showed similar grain size composition containing from 5.48% to 9.83% of the clay fraction (Table 2) and have been classified as sandy loam texture according to the USDA (United States Department of Agriculture) soil classification [32]. In terms of agrotechnical heaviness, they were classified as medium and heavy soils [33].

**Table 2.** Selected physicochemical and biochemical properties of the soil before the application of the distillery stillage (Control in 2018).

| Parameters * | Depth (cm) | Min * | Max * | Me * | CV * |
|---|---|---|---|---|---|
| Clay | 0–20 | 6.07 | 9.83 | 7.00 | 15.35 |
| % | 20–40 | 5.48 | 8.45 | 7.28 | 17.04 |
| pH | 0–20 | 4.42 | 6.37 | 5.83 | 11.78 |
| | 20–40 | 5.03 | 6.58 | 5.74 | 8.42 |
| Corg | 0–20 | 3.9 | 7.7 | 4.45 | 27.93 |
| g kg$^{-1}$ | 20–40 | 1.30 | 5.25 | 3.00 | 40.24 |
| P | 0–20 | 5.59 | 67.80 | 15.30 | 84.62 |
| mg kg$^{-1}$ | 20–40 | 0.18 | 14.98 | 4.35 | 96.00 |
| K | 0–20 | 100 | 243 | 148 | 23.85 |
| mg kg$^{-1}$ | 20–40 | 65.75 | 102 | 95.40 | 14.50 |
| Mg | 0–20 | 68.94 | 98.52 | 74.99 | 80.28 |
| mg kg$^{-1}$ | 20–40 | 40.22 | 68.23 | 56.75 | 16.19 |
| Zn | 0–20 | 1.63 | 3.44 | 2.59 | 19.12 |
| mg kg$^{-1}$ | 20–40 | 1.07 | 3.79 | 1.41 | 55.42 |
| Cu | 0–20 | 0.86 | 2.22 | 1.73 | 28.85 |
| mg kg$^{-1}$ | 20–40 | 0.90 | 5.83 | 1.89 | 69.09 |
| AlP | 0–20 | 0.35 | 1.01 | 0.57 | 33.49 |
| mMpNP kg$^{-1}$ h$^{-1}$ | 20–40 | 0.22 | 0.62 | 0.39 | 29.68 |
| AcP | 0–20 | 1.07 | 1.71 | 1.16 | 17.98 |
| mMpNP kg$^{-1}$ h$^{-1}$ | 20–40 | 0.76 | 1.45 | 1.04 | 21.88 |
| DEH | 0–20 | 0.99 | 1.23 | 1.11 | 6.84 |
| mgTPF kg$^{-1}$ 24 h$^{-1}$ | 20–40 | 0.52 | 0.96 | 0.74 | 22.13 |

* Corg—organic carbon; P—available phosphorus; K—available potassium; Mg available magnesium; Zn—available zinc; Cu—available cooper; AlP—alkaline phosphatase; AcP—acid phosphatase; DEH dehydrogenases; Min—minimum; Max—maximum; Me—median; CV—coefficient of variation.

Before the application of the stillage, the pH of the studied soils was acidic. Exchangeable acidity in the soil level of 0–20 cm ranged from 4.42 to 6.37 pH units, while at the level of 20–40 cm it ranged from 5.03 to 6.58, and the coefficient of variation indicated a low variability of the discussed parameter (Table 2). Soil pH along with the content of soil humus is of the greatest importance for the availability of micro- and macroelements to plants. Under the acidic pH of the soil, the availability and mobility of elements increase. The stillage, considered as acidifying waste, did not have any significant effect on the value of the pH index. After its application, the pH of the analyzed soil did not change radically (Tables 3 and 4) and the pH of most of the analyzed soil samples was still acidic. Only in autumn, in the first year after application, the alkaline reaction was reported in a few soil samples. During this period, a high variation in pH index was found (CV = 21.64%

and CV = 22.08%) (Table 3). The available literature reports that the acidity of the soil may increase immediately after using the stillage. This acidity gradually decreases due to the intense activity of bacteria, which results from the addition of organic matter present in the stillage [12]. This lack of pH change could, therefore, be related to the significant buffering capacity of the organic substance present in the stillage and introduced into the soil with it. According to some authors [34,35], the direction and scope of the influence of exogenous organic matter (EOM) on soil acidification depends on the quantity and quality of the materials used and the properties of the soil itself. Kabaloev et al. [36] found that the use of pure stillage is undesirable because it acidifies the soil, and, therefore, it should be mixed with quicklime to neutralize the acidity. However, the proportion of lime should not exceed 15% of the total weight when mixed with the distiller's waste.

**Table 3.** Selected physicochemical and biochemical properties of the soil in the first year after the application of the stillage (2019).

| Parameters * | Depth cm | Spring | | | | Autumn | | | |
|---|---|---|---|---|---|---|---|---|---|
| | | Min * | Max * | Me * | CV * | Min * | Max * | Me * | CV * |
| pH | 0–20 | 4.50 | 6.15 | 5.69 | 15.16 | 3.97 | 7.06 | 5.31 | 21.64 |
| | 20–40 | 4.58 | 6.44 | 5.73 | 12.60 | 4.06 | 7.14 | 5.11 | 22.08 |
| Corg g kg$^{-1}$ | 0–20 | 5.30 | 12.20 | 8.00 | 27.16 | 9.70 | 15.90 | 11.80 | 15.34 |
| | 20–40 | 3.00 | 5.40 | 4.25 | 22.25 | 3.65 | 7.40 | 5.45 | 22.03 |
| P mg kg$^{-1}$ | 0–20 | 7.12 | 71.83 | 17.43 | 77.04 | 8.09 | 72.14 | 18.09 | 74.18 |
| | 20–40 | 0.18 | 16.12 | 4.83 | 90.33 | 0.57 | 16.98 | 5.05 | 88.99 |
| K mg kg$^{-1}$ | 0–20 | 115 | 259 | 157 | 22.25 | 118 | 267 | 160 | 22.98 |
| | 20–40 | 66.12 | 148 | 5.73 | 23.40 | 67.14 | 152 | 97.00 | 24.15 |
| Mg mg kg$^{-1}$ | 0–20 | 71.62 | 118 | 78.13 | 15.96 | 72.96 | 105 | 79.14 | 11.75 |
| | 20–40 | 41.98 | 69.40 | 58.01 | 14.43 | 42.71 | 68.12 | 60.12 | 15.16 |
| Zn mg kg$^{-1}$ | 0–20 | 0.64 | 3.58 | 2.42 | 36.01 | 2.50 | 6.62 | 2.17 | 25.57 |
| | 20–40 | 0.84 | 1.68 | 1.35 | 24.34 | 1.41 | 2.50 | 2.09 | 21.38 |
| Cu mg kg$^{-1}$ | 0–20 | 0.84 | 1.88 | 1.21 | 24.56 | 1.24 | 1.67 | 1.51 | 9.52 |
| | 20–40 | 0.90 | 1.40 | 1.14 | 16.34 | 1.33 | 1.70 | 1.46 | 8.86 |
| AlP mMpNP kg$^{-1}$ h$^{-1}$ | 0–20 | 0.40 | 1.21 | 0.68 | 34.39 | 0.49 | 1.35 | 0.71 | 34.42 |
| | 20–40 | 0.34 | 0.70 | 0.46 | 24.68 | 0.40 | 0.72 | 0.51 | 23.26 |
| AcP mMpNP kg$^{-1}$ h$^{-1}$ | 0–20 | 1.11 | 1.95 | 1.24 | 20.72 | 1.18 | 2.09 | 1.29 | 21.42 |
| | 20–40 | 0.80 | 1.51 | 1.10 | 22.24 | 0.84 | 1.58 | 1.25 | 20.01 |
| DEH mg TPF kg$^{-1}$ 24 h$^{-1}$ | 0–20 | 1.08 | 1.40 | 1.20 | 7.79 | 1.15 | 1.48 | 1.28 | 6.88 |
| | 20–40 | 0.56 | 1.21 | 0.83 | 26.27 | 0.59 | 1.26 | 0.83 | 26.26 |

* Corg—organic carbon; P—available phosphorus; K—available potassium; Mg—available magnesium; Zn—available zinc; Cu—available cooper; AlP—alkaline phosphatase; AcP—acid phosphatase; DEH—dehydrogenases; Min—minimum; Max—maximum; Me—median; CV—coefficient of variation.

Before the application of the stillage, the amount of Corg ranged from 3.90 to 7.70 g kg$^{-1}$ at the level of 0–20 cm and from 1.30 to 5.25 g·kg$^{-1}$ at the depth of 20–40 cm. According to the criterion adopted in Europe for assessing the content of organic carbon in soils, developed based on the European Soil Database (ESB), these were very low or low values [37]. The low content of organic carbon could have been caused by intensive agricultural cultivation in the analyzed area. The CV coefficients at both studied levels indicated a high variability of Corg. The median value calculated in the distribution analysis suggests that most of the results were below the mean value (Table 2). After using the stillage, a significant increase in the content of organic carbon in the soil was found (Tables 3 and 4) compared to the value of this parameter before the application. The analysis of variance confirmed the significant effect of both the date and depth of sampling on the organic carbon content (Figure 1). Likewise, the increase in the Corg content after the use of various decoctions was noted by other authors [21,38,39]. They found that the stillage, particularly abundant in dry matter (including organic carbon), may be useful in soils with a low organic matter balance. The accumulation of organic matter is mainly related to the soil type. However, the differentiation in the level of this parameter may be related to varied

soil use [40,41]. Intensive agricultural production combined with the simplification of rotation or monoculture may lead to a reduction in the amount of organic residues that enter the cycle of humus transformation, and, consequently, to a decrease in its content in the soil. Humus decomposition and biodegradation may also occur as a result of the use of physiologically acidic fertilizers and the activation of soil microorganisms under the influence of intensive mineral fertilization.

**Table 4.** Selected physicochemical and biochemical properties of the soil in the second year after the application of the stillage (2020).

| Parameters * | Depth cm | Spring | | | | Autumn | | | |
|---|---|---|---|---|---|---|---|---|---|
| | | Min * | Max * | Me * | CV * | Min * | Max * | Me * | CV * |
| pH | 0–20 | 4.32 | 6.04 | 5.78 | 13.6 | 4.14 | 6.04 | 5.13 | 16.64 |
| | 20–40 | 4.21 | 6.45 | 5.65 | 15.13 | 3.80 | 6.40 | 5.26 | 17.83 |
| Corg g kg$^{-1}$ | 0–20 | 9.80 | 16.00 | 11.15 | 16.39 | 9.00 | 1.60 | 10.90 | 17.05 |
| | 20–40 | 4.00 | 8.90 | 8.00 | 5.78 | 4.00 | 6.70 | 5.35 | 18.42 |
| P mg kg$^{-1}$ | 0–20 | 7.12 | 71.83 | 17.43 | 77.83 | 7.25 | 72.69 | 20.95 | 72.01 |
| | 20–40 | 0.58 | 15.59 | 5.09 | 87.13 | 1.02 | 6.32 | 5.44 | 85.14 |
| K mg kg$^{-1}$ | 0–20 | 115 | 259 | 157 | 22.25 | 128 | 296 | 169 | 25.89 |
| | 20–40 | 7.10 | 151 | 93.34 | 24.49 | 45.63 | 109 | 90.73 | 22.73 |
| Mg mg kg$^{-1}$ | 0–20 | 71.6 | 118 | 78.13 | 15.96 | 75.55 | 131 | 77.93 | 19.06 |
| | 20–40 | 27.0 | 66.9 | 56.11 | 33.52 | 6.12 | 70.99 | 54.99 | 46.38 |
| Zn mg kg$^{-1}$ | 0–20 | 1.32 | 3.77 | 2.88 | 32.08 | 2.37 | 3.62 | 3.20 | 15.40 |
| | 20–40 | 1.42 | 3.36 | 2.21 | 29.69 | 2.47 | 3.98 | 3.08 | 15.07 |
| Cu mg kg$^{-1}$ | 0–20 | 1.06 | 1.90 | 1.37 | 17.35 | 0.91 | 1.85 | 1.42 | 23.87 |
| | 20–40 | 1.17 | 1.79 | 1.32 | 15.49 | 1.01 | 1.86 | 1.49 | 23.22 |
| AlP mMpNP kg$^{-1}$ h$^{-1}$ | 0–20 | 0.40 | 1.21 | 0.68 | 34.39 | 0.51 | 1.39 | 0.82 | 31.44 |
| | 20–40 | 0.32 | 0.65 | 0.47 | 23.09 | 0.39 | 0.69 | 0.49 | 19.78 |
| AcP mMpNP kg$^{-1}$ h$^{-1}$ | 0–20 | 1.11 | 1.95 | 1.24 | 20.72 | 1.19 | 2.25 | 1.33 | 22.46 |
| | 20–40 | 0.81 | 1.51 | 1.12 | 20.67 | 0.96 | 1.63 | 1.21 | 16.41 |
| DEH mg TPF kg$^{-1}$ 24 h$^{-1}$ | 0–20 | 1.09 | 1.40 | 1.20 | 7.79 | 1.16 | 1.51 | 1.23 | 8.931 |
| | 20–40 | 0.53 | 1.22 | 0.81 | 26.64 | 0.64 | 1.38 | 0.90 | 24.12 |

* Corg—organic carbon; P—available phosphorus; K—available potassium; Mg—available magnesium; Zn—available zinc; Cu—available cooper; AlP—alkaline phosphatase; AcP—acid phosphatase; DEH—dehydrogenases; Min—minimum; Max—maximum; Me—median; CV—coefficient of variation.

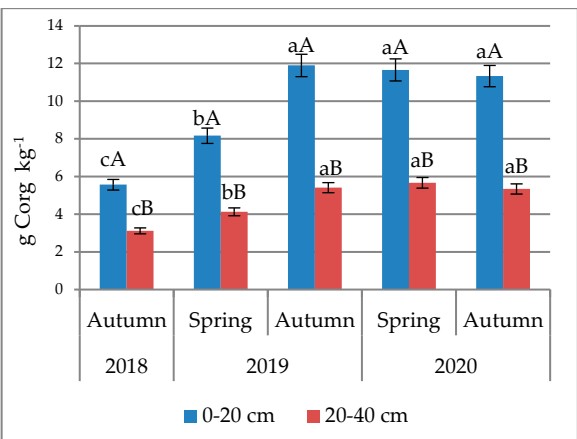

**Figure 1.** Content of organic carbon (Corg) in the studied soil. Different lower-case letters indicate significant differences among the soil sampling seasons (2018 autumn, 2019 and 2020 spring, autumn). Different upper-case letters indicate significant differences among levels of sampling. Error bars represent standard deviation.

The P content in the soil before the application of the distillery stillage, at the level of 0–20 cm, ranged from 5.59 mg kg$^{-1}$ to 67.80 mg kg$^{-1}$ (Table 2). On the other hand, at the level of 20–40 cm, the content was lower, from 0.177 mg kg$^{-1}$ to 14.98 mg kg$^{-1}$. The

analysis of variance showed a significant effect of the timing and depth of soil sampling on the content of available P (Figure 2A). The highest P content was found in the spring of 2020 (the second year after stillage application), from the level of 0–20 cm (26.29 mg kg$^{-1}$). Based on the average content, the tested soil can be classified as low and very low content of available P (4th and 5th class), according to the norm PN-R-04023 [23]. The content of available phosphorus at ~30 mg kg$^{-1}$ of soil can be considered a critical value for plants; therefore, it is recommended to intensify fertilization with mineral phosphorus. As for the level of 20–40 cm, the highest content of P (7.51 mg kg$^{-1}$) was found in soil samples collected in autumn 2020. No significant differences were found between the P value in the soil sampled in autumn 2019 and spring 2020, and the fall of 2019 (control) and the spring of 2020. Significantly higher P content was found at the level of 0–20 cm. The content of available phosphorus decreased significantly in the deeper soil level (20–40 cm). Phosphorus is a non-mobile element in the soil and usually remains in the layer to which it was introduced with the fertilizer [42]. Two years after the application of the stillage, the P content in the soil increased by ca. 23%. Available P is characterized by a low utilization rate in the first year after fertilization. CV values for P ranged from 72.01 to 84.62% (0–20 cm) and 85.14–96.00% (20–40 cm), throughout the experiment, indicating a high variability of this parameter. However, the distribution analysis showed that most of the results are below the average, as indicated by the median values lower than the mean (Tables 2–4).

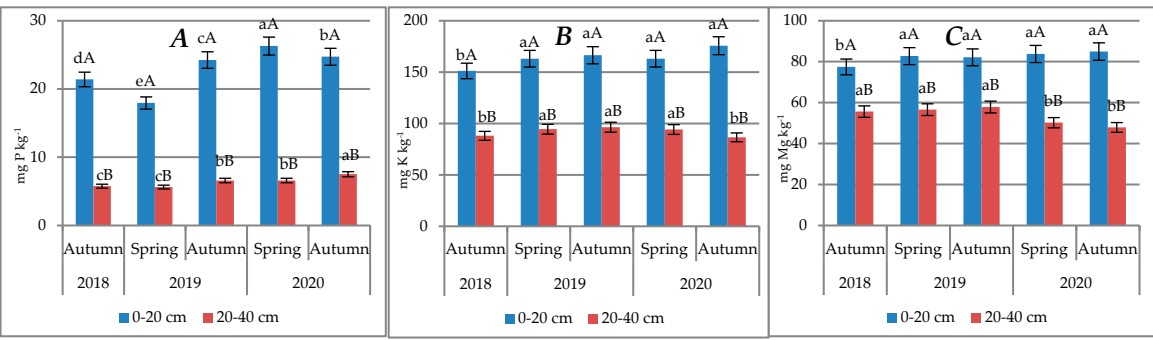

**Figure 2.** (**A**–**C**). Contents of available P (**A**), K (**B**), and Mg (**C**) in the studied soil. Different lower-case letters indicate significant differences among the soil sampling seasons (2018 autumn, 2019 and 2020 spring, autumn). Different upper-case letters indicate significant differences among the levels of sampling. Error bars represent standard deviation.

The analysis of variance (ANOVA) showed a significant dependence of the K content on the date and depth of soil sampling. The highest content of available K was found in soil samples collected in autumn 2020 (175 mg kg$^{-1}$) at the level of 0–20 cm (Figure 2B). This means an increase of 16% compared to the control (151 mg kg$^{-1}$). According to the PN-R-04022 standard [26], the tested soil is of 3rd class with a medium content of K. The amount of available K was significantly lower at the level of 20–40 cm. As for absorbable K, the CV value ranged from 11.75% to 23.85% (0–20 cm) (Tables 1–3), indicating a low and moderate variability in its content. According to Skowrońska and Filipek [6], a significant part of P and K introduced with the stillage remains in the pool readily available to the plants. This is probably why there was an increase in the content of available P and K forms found in the soil fertilized with the distillery stillage. According to Szulc et al. [3], a common feature of stillages (produced from rye, potato, and molasses) is that the content of phosphorus is too low to nitrogen and potassium. Therefore, the fertilizing use of stillages also requires the consideration of mineral fertilization.

The content of Mg in the soil was significantly diverse in the individual years of the study, as well as in the levels of soil sampling. There were no significant differences between the seasons of sampling in both years after the application of the stillage (Figure 2C). The lowest Mg content was reported at the level of 0–20 cm in the fall before the application of the stillage (77.42 mg kg$^{-1}$). According to the PN-R-04020 [27] standard, the level of

available Mg was medium, which classifies the soil studied into the 3rd class of abundance in this element. The content of available Mg before the start of the research ranged from 68.94 to 98.52 mg kg$^{-1}$ (0–20 cm) and from 40.22 to 68.23 mg kg$^{-1}$ (20–40 cm) (Table 2). In the following years, the content of this macronutrient increased (Tables 3 and 4). In all terms of soil sampling, the median value was lower than the average content of this element, which indicates that most of the results are lower than the average Mg content in the soil samples.

The content of available Zn and Cu micronutrients in the soil before the application of the stillage was low (Table 2). The comparison of the content of the extracted zinc DTPA in the studied soils with the critical values for arable soils shows that our soil samples were characterized by a higher than the deficit content of available Zn [28]. Moreover, available forms of Cu were present in amounts above the deficit values, which were assumed as 0.12–0.25 mg kg$^{-1}$ [43]. It should be emphasized that the amounts of this metal in the soil in a phyto-available form were generally low, ranging from 0.89 to 5.83 mg kg$^{-1}$. The distillery stillage with a low content of trace elements did not significantly affect the amount of available Zn and Cu in the analyzed soil. After the application of the stillage, the contents of Zn and Cu forms extracted with DTPA were still low and reached for Zn: 1.31–2.35 mg kg$^{-1}$ (in the first year after application) and 2.33–3.20 mg kg$^{-1}$ (in the second year), and for Cu: 1.10–1.48 mg kg$^{-1}$ (in the first year) and 1.32–1.42 mg kg$^{-1}$ (the second year after application) (Tables 3 and 4). The probable reasons for such low content of the analyzed microelements may be the lack of their supplementation in the fertilizers used, as well as their depletion from the soil as a result of many years of agricultural activity, including monoculture. Literature data [5,21] describe the stillage as poor in microelements, including heavy metals, which do not exceed the limit values established for waste used following the recovery process. When studying the effect of stillage on soil properties, Ramalho et al. [44] did not notice a significant increase in the content of micronutrients. They also found that there may be a slight risk of soil contamination with heavy metals when using different types of distillery stillages. The mobility of nutrients in the soil depends on its granulometric composition, the content of organic matter, pH, and microbiological activity. The method of soil use and the applied mineral and organic fertilization are also meaningful [38,45]. The differentiation in the uptake of micronutrients by plants is particularly high under acidic soil conditions when their solubility is increased.

The differences in the assimilable Zn content between the compared sampling levels were significant at all times tested, except for autumn 2020. The highest Zn content was found in autumn 2018 and 2020 at the surface level (Figure 3A). As for the available Cu, the highest content was found in autumn 2018 at the subsurface level and it was significantly higher compared to all other dates. Significant differences between the compared depths were found only on two dates of sampling, i.e., autumn 2018 and 2020 (Figure 3B).

Soil enzymes are often used to assess changes occurring in the soil environment under the influence of chemical substances [19,45]. In the present study, it was found that both the timing and depth of soil sampling had a significant effect on the activity of AlP and AcP (Figure 4A,B). The highest activity of both phosphatases was obtained at the level of 0–20 cm. The highest activity of AlP (0.85 mMpNP kg$^{-1}$ h$^{-1}$) was found in the soil sampled in autumn 2020, i.e., in the second year after the application of the stillage. Its activity increased by 36% compared to the control (autumn 2018). AlP activity at the depth of 0–20 cm (control; autumn 2018) ranged from 0.35 to 1.01 mMpNP kg$^{-1}$ h$^{-1}$ (Table 2). This range indicates a fairly high variability of the obtained results. The observed differentiation may be confirmed by the high value of the coefficient of variation (33.49%). On the other hand, the analysis of distribution showed that most of the results are below the mean value, confirmed by the low median (0.57) value. In the following years, the CV value was at a similar level after the application of the distillery stillage (Tables 3 and 4). The highest activity of AcP was reported in the fall in the first and second years after the application of the stillage (Figure 4B). The activity of AcP was higher than that of AlP, which was related to the acidic reaction of the soil. This higher activity of acid phosphatase, reported

in the tested soil samples, can be explained by the fact that phosphomonoesterases are very sensitive to changes in the soil pH [46,47], which in this case was acidic. These conditions were optimal for AcP activity.

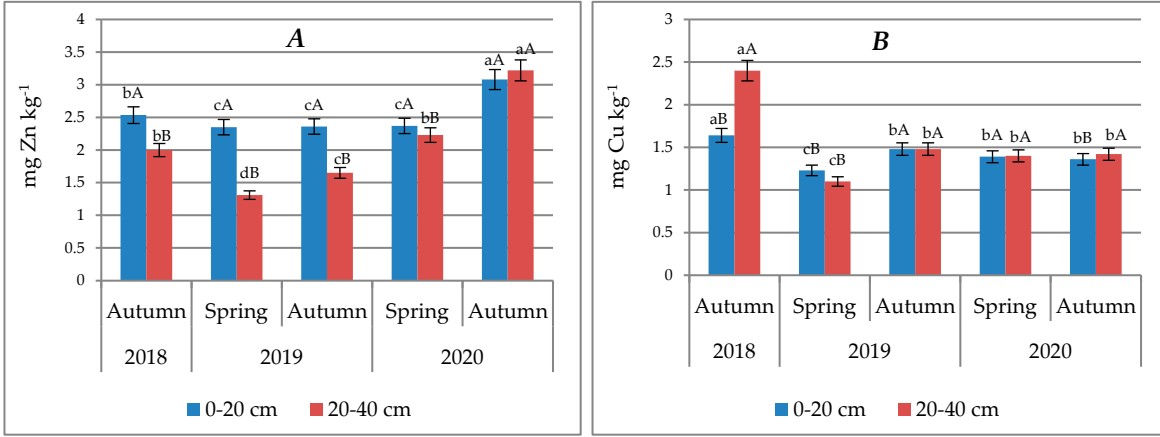

**Figure 3.** (**A**,**B**). Contents of available Cu (**A**) and Zn (**B**) in the studied soil. Different lower-case letters indicate significant differences among the soil sampling seasons (2018 autumn, 2019 and 2020 spring, autumn). Different upper-case letters indicate significant differences among the levels of sampling. Error bars represent standard deviation.

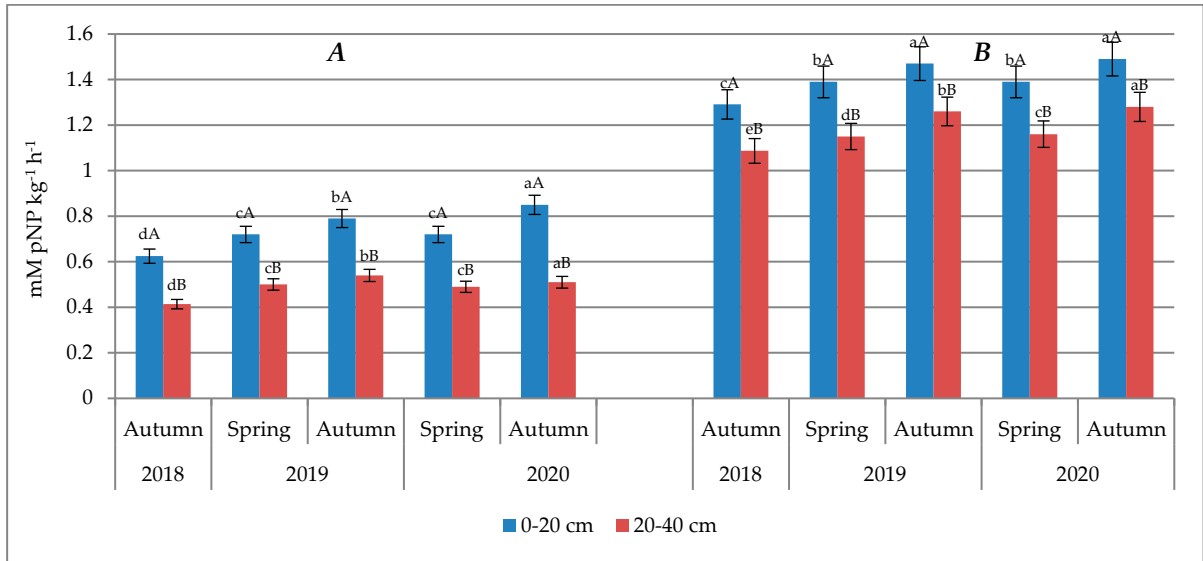

**Figure 4.** (**A**,**B**). Activity of alkaline (AlP) (**A**) and acid (AcP) (**B**) phosphatases in the studied soil. Different lower-case letters indicate significant differences among the soil sampling seasons (2018 autumn, 2019 and 2020 spring, autumn). Different upper-case letters indicate significant differences among the levels of sampling. Error bars represent standard deviation.

Dehydrogenases belong to the class of oxidoreductases. They catalyze the oxidation of organic compounds by releasing electrons and protons from them. The redox potential influences the biochemical processes in the soil by affecting the growth and development of microorganisms that are the source of soil enzymes [18,48]. The ANOVA confirmed a significant effect of both the timing and level of soil sampling. The highest DEH activity was reported in autumn 2019 (1.30 mg TPF kg$^{-1}$ 24 h$^{-1}$) and 2020 (1.27 mg TPF kg$^{-1}$ 24 h$^{-1}$) from the level $0-20$ cm (Figure 5). There were no significant differences between the spring dates. The lowest DEH activity was found in the soil without the application of stillage (autumn of 2018). Similarly, the studies by Kaur et al. [20] showed an increase in

DEH activity after the application of the stillage, which the authors explain by an increase in the content of organic matter in the soil. The activity of AlP, AcP, and DEH was higher in the soil at a depth of 0–20 cm. Usually, the enzyme activity is higher in the soil surface and decreases at deeper levels. This can be explained by the profile content of organic matter [49].

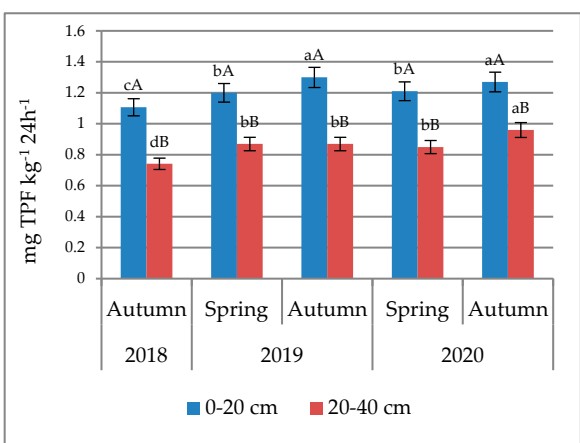

**Figure 5.** Activity of dehydrogenases (DEH) in the studied soil. Different lower-case letters indicate significant differences among the soil sampling seasons (2018 autumn, 2019 and 2020 spring, autumn). Different upper-case letters indicate significant differences among the levels of sampling. Error bars represent standard deviation.

The relationships between the soil parameters (pH in KCl, Corg, P, K, Mg, Cu, Zn, DEH, AlP, and AcP) and the application of the stillage were investigated by using the multivariate PCA (principal components method).

The PCA for the control (2018) (Figure 6) showed that the first component (PC 1) that generated 54.06% of the total variance has the following factor loadings for the individual characteristics: $PC1 = -0.013_{clay} + 0.090_{pH} - 0.968_{Corg} - 0.905_P - 0.767_K - 0.976_{Mg} - 0.378_{Zn} + 0.326_{Cu} - 0.937_{AlP} - 0.823_{AcP} - 0.912_{DEH}$. The second component (PC2), which explained 16.33% of the total variance, has the following loadings for each of the traits: $PC2 = -0.613_{clay} - 0.115_{pH} - 0.055_{Corg} - 0.088_P - 0.180_K - 0.019_{Mg} - 0.707_{Zn} - 0.908_{Cu} - 0.008_{AlP} - 0.137_{AcP} - 0.128_{DEH}$.

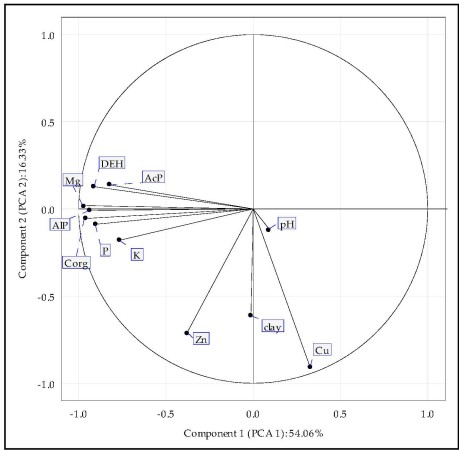

**Figure 6.** Configuration of variables in the system of the first two axes PC1 and PC2 of principal components in the control soil in 2018.

The conducted PCA analysis for 2019 (Figure 7A) indicated that the first component (PC1) generated 54.26% of the total variance and it was mainly negatively related to the

analyzed parameters: PC1 = $-0.098_{clay} - 0.162_{pH} - 0.807_{Corg} - 0.912_P - 0.778_K - 0.950_{Mg} - 0.685_{Zn} - 0.459_{Cu} - 0.907_{AlP} - 0.815_{AcP} - 0.877_{DEH}$. Most of the significant values of the loadings mean that the greater the intensity of these variables, the more important is their contribution to PC1. Conversely, PC2 generated 12.53% of the total variance and was positively associated with clay (0.778) and pH (0.820).

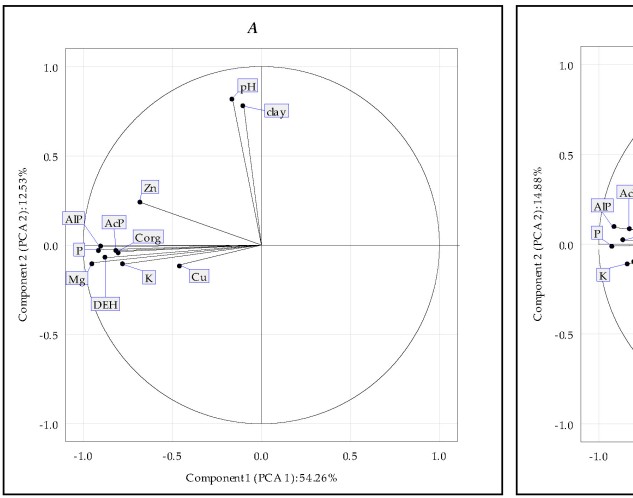
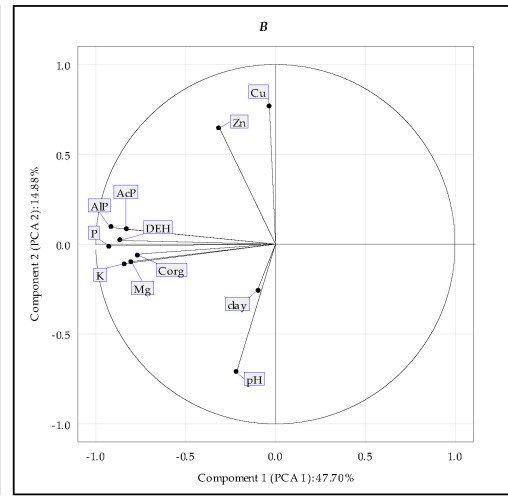

**Figure 7.** (**A**,**B**). Configuration of variables in the system of the first two axes PC1 and PC2 of principal components in the soil after application distillery stillage A (2019 year) and B (2020 year).

Additionally, in 2020, the results of PCA (Figure 7B) were similar. The first component represented 47.70% of the total variance. PC1 has been associated with PC1 = $-0.102_{clay} - 0.217_{pH} - 0.775_{Corg} - 0.925_P - 0.839_K - 0.802_{Mg} - 0.311_{Zn} - 0.033_{Cu} - 0.920_{AlP} - 0.833_{AcP} - 0.862_{DEH}$. The PC2, explaining the 14.88% of the variation, was negatively correlated with the pH (0.713) and positively with Cu content (0.776). The presented PCA showed that most of the analyzed soil parameters (the content of available macroelements and soil biochemical activity) were grouped on one side of the PC1 factor axis, which, therefore, can be explained by the influence of stillage in the soil. The second component was mainly a measure of the level of micronutrients (Cu and Zn) in the soil, since the first eigenvector shows approximately equal negative loadings on all variables.

The PCA also allowed us to verify the importance of the mutual correlations between the studied parameters. The PCA correlation analysis showed a significant relationship between the content of Corg and P ($r = 0.831$), K ($r = 0.660$), Mg ($r = 0.945$) and the activity of AlP ($r = 0.931$), AcP ($r = 0.812$) and DEH ($r = 0.895$) in the soil prior to the application of the distillery stillage (Figure 6). Usually, the DEH activity has a close relationship with the activity of other enzymes, mainly AlP and AcP, indicated by high correlations between these parameters.

After analyzing the distribution of basic soil properties and the amount of extracted trace elements on the plot of the factor coordinates (PCA), it was found that the available forms of zinc were strongly associated with the soil pH in the first year after the application of the stillage (Figure 7A). Additionally, in the first year after using the stillage, a significant correlation was found between the content of Corg and P ($r = 0.606$), K ($r = 0.627$), Mg ($r = 0.764$) and the activity of AlP ($r = 0.561$), AcP ($r = 0.426$) and DEH ($r = 0.626$) (Figure 7A). According to Rusecki et al. [50], the content of organic matter in the soil is involved in the chelation of cations that limit the absorption of P and increases the solubility of P compounds in the soil. However, the values of the correlation coefficient were lower than in the soil before the stillage application. In the present study, a relationship was found between the content of available forms of Zn and Cu, and the content of Corg in the first year of research (Figure 7A). The graph shows that the available zinc content was positively correlated with the organic carbon ($r = 0.561$), determined in the first year

of the experiment, and the content of phosphorus ($r$ = 0.505), potassium ($r$ = 0.421), and magnesium ($r$ = 0.551). Organic matter is the basic adsorbent of trace elements under acidic conditions and, thus, limits their availability to the plants [39]. A significant positive correlation was found between the Zn content in the soil and the activity of AlP ($r$ = 0.567), AcP ($r$ = 0.518), and DEH ($r$ = 0.510), and between Cu and AlP ($r$ = 0.348) and AcP ($r$ = 0.342). The activity of some enzymes depends on metal ions as they act as cofactors. Heavy metals in low concentrations can be enzyme activators in the soils [18]. In the present study, their acceptable content was not exceeded, therefore, probably no inhibition of AlP, AcP, and DEH activity occurred.

A significant positive correlation was found between the content of available P and AlP, AcP in the soil at all dates of soil sampling (Figures 6 and 7A,B). Soil phosphatases catalyze the hydrolysis of organic phosphorus compounds and are used to assess the potential rate of mineralization of these compounds in the soil. Organophosphorus compounds found in soil are the substrate for phosphomonesterases [46]. Knowledge of the extent of these two parameters should enable the estimation of the content of phosphorus available for plants, which can be determined by the Egner-Riehm method.

## 4. Conclusions

The results obtained from the three-year experiment showed that fertilization with distillery stillage had a positive effect on soil properties. The acidic pH of the stillage had little effect on lowering the pH of the studied soil. However, systematic monitoring and regulation of soil pH are necessary. The use of stillage as fertilizer had a significant effect on the content of organic carbon in the analyzed soil. As for P, K, and Mg, their content increased during the research and in both soil levels studied. The average content of available macronutrients allowed us to classify the soil into the 3rd class of fertility. Therefore, it is advisable to use mineral fertilization simultaneously. The activity of the tested enzymes increased after the application of rye stillage, which underlines its positive effect on soil fertility. The use of distillery stillage favors the recycling of minerals in the agricultural sector, which, in turn, brings environmental benefits and may contribute to the reduction in costs related to the application of mineral fertilizers.

**Author Contributions:** Conceptualization, A.B. and J.L.; methodology, J.L. and A.B.; software, A.B. and J.L.; validation, J.L.; A.B and M.R.; formal analysis, A.B., J.L., M.R., O.D. and K.E.; investigation, J.L., A.B., M.R., K.E. and O.D.; resources, A.B. and J.L.; data curation, A.B. and J.L.; writing—original draft preparation, A.B. and J.L.; writing—review and editing, J.L. and A.B.; visualization, J.L., A.B. and M.R.; supervision, A.B. and J.L.; project administration, J.L. and A.B. All authors have read and agreed to the published version of the manuscript.

**Funding:** Bydgoszcz University of Science and Technology under Grant BN 38/2019.

**Institutional Review Board Statement:** Not applicable.

**Informed Consent Statement:** Not applicable.

**Data Availability Statement:** Not applicable.

**Acknowledgments:** The authors would like to thank the Faculty of Agriculture and Biotechnology, Bydgoszcz University of Science and Technology for their support in this research work.

**Conflicts of Interest:** The authors declare no conflict of interest.

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
