# Peer review of "Enzymatic Activity of Soil after Applications Distillery Stillage"

_agriculture, doi:10.3390/agriculture12050652_

Round 1

Reviewer 1 Report

The present study investigate the effect of stillage obtained after rye processing on the changes in chemical parameters and enzymatic activity during the three years of the research. The authors have to improve some parts .

  1. Highlight the novelties of your results in the foreground and compared your results with the relevant recent literature in the Result and discussion.
  2. Table 3 and 4 are missing one parameter (Cly %) regards to Table 2.

Author Response

Dear Reviewer

Thank you for your thorough review of the manuscript and constructive criticism  Bartkowiak A., Lemanowicz J., Rydlewska M., DrabiÅ„ska O., Ewert K.  “Enzymatic activity of soil after application distillery stillage”.

 We agree with the comments of the Reviewers and so the article has been corrected and supplemented compliant with the  suggestions. We hope that all the corrections required by the Reviewers have been adequately made.

  1. In response to your suggestions, the news of our results has been highlighted in Introduction and our results have been compared with the relevant recent literature in the Results and Discussion.
  2. Table 3 and 4 are missing one parameter (Clay %) regards to Table 2.

This is explained on lines 166 - 168.

“As the soil texture is a parameter that does not undergo seasonal changes, the content of the loam fraction is presented only in Table 2. It remained unchanged for the rest of the time.”

Kind Regards

Agata Bartkowiak

Reviewer 2 Report

Overall, this version is well. But the presentation is not adequate and concise,  in particular, table 3 and 4 are somewhat redundant when comparing to following text.

Author Response

Dear Ediror

Agriculture

Thank you for your thorough review of the manuscript and constructive criticism  Bartkowiak A., Lemanowicz J., Rydlewska M., DrabiÅ„ska O., Ewert K.  “Enzymatic activity of soil after application distillery stillage”.

 We agree with the comments of the Reviewers and so the article has been corrected and supplemented compliant with the  suggestions. We hope that all the corrections required by the Reviewers have been adequately made.

The answers to the comments of Reviewer 1.

  1. In response to your suggestions, the news of our results has been highlighted in Introduction and our results have been compared with the relevant recent literature in the Results and Discussion.
  2. Table 3 and 4 are missing one parameter (Clay %) regards to Table 2.

This is explained on lines 166 - 168.

“As the soil texture is a parameter that does not undergo seasonal changes, the content of the loam fraction is presented only in Table 2. It remained unchanged for the rest of the time.”

The answers to the comments of Reviewer 2.

1.Overall, this version is well. But the presentation is not adequate and concise,  in particular, table 3 and 4 are somewhat redundant when comparing to following text.

Answer: Tables 3 and 4 are necessary because they show changes to the  selected physicochemical and biochemical properties of the soil in the first and second year after the stillage application.

The answers to the comments of Reviewer 3.

  1. The stillage was used only once (in 2018, before the soil samples were collected). Control samples (prior to the use of stillage) were sampled in autumn 2018. 100 soil samples were sampled over the entire experimental period (2018-2022). Soil samples were collected from surface mineral levels from depths of 0–20 cm and 20-40 cm. At each period, 40 soil samples were collected (20 soil samples in the depths 0-20 cm and 20 soil samples in the depths 20-40 cm. Soil samples were collected by the squares method, from the plot of 2 ha. The average sample consisted of ten primary samples.

2.In response to your suggestions, changes were made to the Abstract section. The aim of our study was not carbon sequestration, therefore we are not discussing this topic.

  1. After consulting all the manuscript co-authors , we found that the tables and figures remain unchanged. I clearly show the changes of the examined soil parameters before and after the application of the decoction.

The answers to the comments of Reviewer 4.

  1. The title of the manuscript was slightly changed.
  2. Changes to the Abstract section have been made.
  3. In the Materials and methods section:

It was described how many soil samples were taken and the description of the statistical analysis (ANOVA and PCA) was changed.

4.In the Introduction section:

- Added information about soil enzymes. The purpose of the manuscript  has also been improved.

  1. In the Results and Discussion section:

- The contents of Tables 2-4 are not  repeated with Figs 1-5. Tables 2-4 include parameters  min, max, median and cv, while the figures 1-5 are based on averages.

- Error bars represent standard deviation. The graphs now has a same annotation.

- "levels of sampling" mean the depth of the soil.

  1. In the References section deleted some references.

The manuscript was reviewed by a native English-language reviewer Tim Brombley (language editor of the Geography Bulletin. Physical Physical Series).

We hope that the corrections made meet the requirements of the Journal and so we could expect our article being published at your earliest convenience.

Kind Regards

Agata Bartkowiak

Reviewer 3 Report

Review report Enzymatic Activity of Luvisoils After Using Distillery Stillage

The paper deal with an interesting topic and is of interest to the readers of Agronomy journal. However, I have identified some critical weaknesses in the experimental setup that I list below.

Main weakness:

I think that the main weakness of this work is the lack of a control plot during the monitoring period and the number of samples processed. No control parcel was monitored during the sampling period to have a comparison with. The control was sampled only before stillage application.

Moreover, from what I understand, real replicates are missing, and statistic is performed basing on three analysis made on the same composite sample. The field should have been parcelled to allow comparison and statistic should have been made on samples coming from different plots.

The number of samples is extremely low considering the long experimental period. From what I understand, 6 average samples were taken in total over the whole period (1 pre stillage application, 1 post stillage in Oct 2018, 2 in 2019 and 2 in 2020) at 2 depths.

This makes the study not sufficiently robust from my point of view for a research article especially because of the lack of real replicates and controls during the monitored period and comparison with other agronomical practices to address the contribution brought by stillage.

Some comments that may be useful for the authors:

I am not a native English speaker, so I cannot undergo revision of the English style and grammar of the manuscript. However, I am afraid that the English quality is not good enough for a scientific publication. I suggest the authors use more adequate terminology and have the manuscript reviewed by a native English speaker. 

Abstract:

“using the methods common in soil science laboratories” à this statement is not necessary.

“The activities of the enzymes: 14 alkaline phosphatase (AlP), acid phosphatase (AcP), and dehydrogenases (DEH), were also tested” à please rephrase: “Activities of ……. Enzymes were also accounted” ore something similar.

“season of testing” à this terminology is a bit confusing: do you mean seasonal effects (different enzyme activity to sampling period) or different enzyme activity to application period (you applied stillage in different periods and you measured differences?). I suppose the first. Please rephrase accordingly

It is not clear why you state that your treatment does not affect pH, but then you state that you need to regulate pH. For sure you will explain this in the discussion, but while reading the abstract this is confusing. At least state why you need to regulate pH if your treatment is not affecting soil pH.

Introduction

Line 50: modify with “compared to” Nitrogen and Potassium.

Line 55: you state that biodegradation of organic acids produces CO2 and H2O. I would expect that the addition of this material to soil thus enhances soil respiration and CO2 emissions with eventually some negative impacts in the prospect of climate change. Is there any study in the scientific literature that addresses this aspect? If so, it would be great to also include a discussion on this.

Material and Methods

The modality by which the stillage was applied is not clear.

You state this: “The average sample consisted of ten primary samples. The distillery stillage was applied in October 2018 at a dose of 40,000 litres per hectare for the cultivation of winter triticale cv. ‘Grenado’ in monoculture. Soil samples were taken before the use of the stillage (control in 2018) and seasonally (spring, autumn) in the following two years (2019 and 2020) after applying the stillage as a fertilizer”

But honestly, I cannot understand later in the MS graphs if the “Autumn 2018” then refers to the pre stillage application or if it is already after the stillage application (the first application). So, are the pre-fertilization values shown in the graph? Am I missing something?

Moreover, it is not clear if the 40,000 litres/ha were applied once in autumn 2018 or if stillage was applied (I suppose at the same amount) also in spring/autumn (both?) in 2019 and 2020.

On which base do you apply 40,000 litres/ha? N content?

Consideration of results and discussions and data presentation:

Tables: I suggest changing the table format, it is hard to compare the results of pre-fertilization with post-fertilization.

Suggestion: Instead of showing the descriptive statistic, you could eventually show in a single table the results from each sampling period (i.e. before stillage and after as mean ± st. dev with letters from statistic). You can eventually also divide enzyme activity from other chemical parameters. You can build different tables for different soil layers if there is not enough space, since it seems that you have a clear difference between 0-20 and 20-40 soil layers.

Table 2: “cly” I suppose should be “Clay”

Are decimals for numbers > 100 significant?

Author Response

Bydgoszcz, on this day of 11.04.2022

Dear Editor

Agriculture

Thank you for your thorough review of the manuscript and constructive criticism  Bartkowiak A., Lemanowicz J., Rydlewska M., DrabiÅ„ska O., Ewert K.  “Enzymatic activity of soil after application distillery stillage”.

 We agree with the comments of the Reviewers and so the article has been corrected and supplemented compliant with the  suggestions. We hope that all the corrections required by the Reviewers have been adequately made.

The answers to the comments of Reviewer 3.

  1. The stillage was used only once (in 2018, before the soil samples were collected). Control samples (prior to the use of stillage) were sampled in autumn 2018. 100 soil samples were sampled over the entire experimental period (2018-2022). Soil samples were collected from surface mineral levels from depths of 0–20 cm and 20-40 cm. At each period, 40 soil samples were collected (20 soil samples in the depths 0-20 cm and 20 soil samples in the depths 20-40 cm. Soil samples were collected by the squares method, from the plot of 2 ha. The average sample consisted of ten primary samples.

2.In response to your suggestions, changes were made to the Abstract section. The aim of our study was not carbon sequestration, therefore we are not discussing this topic.

  1. After consulting all the manuscript co-authors , we found that the tables and figures remain unchanged. I clearly show the changes of the examined soil parameters before and after the application of the decoction.

Kind Regards

Agata Bartkowiak

Reviewer 4 Report

Title: Add “soil”

Abstract:

L17-18: No change in pH? Please try to use: ... has no significant effect on pH

Introduction:

Add information about soil enzymes. The authors didn't say why they were doing soil enzymes. The research purpose does not correspond to the title.

Materials and Methods

How many soil samples were taken?

Statistical analysis: There should be an analysis of the treatment effects in ANOVA. Different sampling time belongs to a repeated measures ANOVA. The purpose of PCA needs to be added.

Results and Discussion

The contents of Tables 2-4 are repeated with Figs 1-5!

Each graph has a different annotation. What does the error bar represent?

L215-216: What does "levels of sampling" mean? Depth of soil?

The discussion is not thorough enough and the main points of the paper should be highlighted.

There are too many references, and please delete some that are irrelevant to this paper.

Author Response

Bydgoszcz, on this day of 11.04.2022

Dear Editor

Agriculture

Thank you for your thorough review of the manuscript and constructive criticism  Bartkowiak A., Lemanowicz J., Rydlewska M., DrabiÅ„ska O., Ewert K.  “Enzymatic activity of soil after application distillery stillage”.

 We agree with the comments of the Reviewers and so the article has been corrected and supplemented compliant with the  suggestions. We hope that all the corrections required by the Reviewers have been adequately made.

The answers to the comments of Reviewer 4.

  1. The title of the manuscript was slightly changed.
  2. Changes to the Abstract section have been made.
  3. In the Materials and methods section:

It was described how many soil samples were taken and the description of the statistical analysis (ANOVA and PCA) was changed.

4.In the Introduction section:

- Added information about soil enzymes. The purpose of the manuscript  has also been improved.

  1. In the Results and Discussion section:

- The contents of Tables 2-4 are not  repeated with Figs 1-5. Tables 2-4 include parameters  min, max, median and cv, while the figures 1-5 are based on averages.

- Error bars represent standard deviation. The graphs now has a same annotation.

- "levels of sampling" mean the depth of the soil.

  1. In the References section deleted some references.

The manuscript was reviewed by a native English-language reviewer Tim Brombley (language editor of the Geography Bulletin. Physical Physical Series).

We hope that the corrections made meet the requirements of the Journal and so we could expect our article being published at your earliest convenience.

Kind Regards

Agata Bartkowiak

Round 2

Reviewer 3 Report

I have no other suggestions for authors

Author Response

Dear Reviewer

Agriculture

Thank you for your next insightful review of the manuscript Bartkowiak A., Lemanowicz J., Rydlewska M., DrabiÅ„ska O., Ewert K. “Enzymatic Activity of Soils After Using Applications Distillery Stillage”.

The article has been corrected according to the suggestions. We hope that all the required corrections have been adequately implemented.

Kind Regards

Agata Bartkowiak

Reviewer 4 Report

Title: Delete extra spaces. “soil” should begin with a capital letter.

There are extra spaces in other parts of the article. Please check them carefully and delete them.

Abstract:

L15:copper (Cu).

L15-16: grammatically wrong sentences: “Enzymes were also accounted: alkaline phosphatase (AlP), acid phosphatase (AcP), and dehydrogenases (DEH), were also tested.” 

L18: “and did not has no significant effect on pH”. This is a grammatically wrong sentence.

L20: it is necessary to systematically monitor the soil pH

Introduction:

The first and second paragraphs about stillage are too detailed and could be further condensed.

L76: From here, soil enzymes are introduced, which are different from C and nutrients mentioned above. So it's recommended to start another paragraph here.

Results and Discussion

Table 2-4: the max of K ?? Are there any typing problems? So does the number in 20-40cm soil layer of K. Please pay attention to the significant digits.

Author Response

Dear Reviewer

Thank you for your next insightful review of the manuscript Bartkowiak A., Lemanowicz J., Rydlewska M., DrabiÅ„ska O., Ewert K. “Enzymatic Activity of Soils After Using Applications Distillery Stillage”.

The article has been corrected according to the suggestions. We hope that all the required corrections have been adequately implemented.

The sentences have been stylistically corrected.

Max contents of K included in tables 2-4 are correct. This is not an error and there are no typing errors as well.

Kind Regards

Agata Bartkowiak